# Personalized Treatment for Infantile Ascending Hereditary Spastic Paralysis Based on In Silico Strategies

**DOI:** 10.3390/molecules27207063

**Published:** 2022-10-19

**Authors:** Matteo Rossi Sebastiano, Giuseppe Ermondi, Kai Sato, Asako Otomo, Shinji Hadano, Giulia Caron

**Affiliations:** 1Molecular Biotechnology and Health Sciences Department, University of Torino, Quarello 15, 10135 Torino, Italy; 2Molecular Neuropathobiology Laboratory, Department of Molecular Life Sciences, Tokai University School of Medicine, 143 Shimokasuya, Isehara 259-1193, Japan

**Keywords:** infantile onset ascending hereditary spastic paralysis, ALS2, virtual screening, vitamin K, drug repurposing, personalized medicine

## Abstract

Infantile onset hereditary spastic paralysis (IAHSP) is a rare neurological disease diagnosed in less than 50 children worldwide. It is transmitted with a recessive pattern and originates from mutations of the *ALS2* gene, encoding for the protein alsin and involved in differentiation and maintenance of the upper motoneuron. The exact pathogenic mechanisms of IAHSP and other neurodevelopmental diseases are still largely unknown. However, previous studies revealed that, in the cytosolic compartment, alsin is present as an active tetramer, first assembled from dimer pairs. The C-terminal VPS9 domain is a key interaction site for alsin dimerization. Here, we present an innovative drug discovery strategy, which identified a drug candidate to potentially treat a patient harboring two *ALS2* mutations: one truncation at lysine 1457 (not considered) and the substitution of arginine 1611 with a tryptophan (R1611W) in the C-terminus VPS9. With a protein modeling approach, we obtained a R1611W mutant model and characterized the impact of the mutation on the stability and flexibility of VPS9. Furthermore, we showed how arginine 1611 is essential for alsin’s homo-dimerization and how, when mutated to tryptophan, it leads to an abnormal dimerization pattern, disrupting the formation of active tetramers. Finally, we performed a virtual screening, individuating an already therapy-approved compound (MK4) able to mask the mutant residue and re-establishing the alsin tetramers in HeLa cells. MK4 has now been approved for compassionate use.

## 1. Introduction

Infantile-onset ascending hereditary spastic paralysis (IAHSP) is a rare, early-onset autosomal recessive motor neuron disease associated with mutations in the *ALS2* gene [1], which is also responsible for other motor neuron diseases (MNDs), such as ALS type II [2] and juvenile primary lateral sclerosis (JPLS) [3]. IAHSP affects less than 50 people worldwide. The pathogenesis starts in the early childhood, with a progressive degeneration of upper motor neurons, progressively hindering deambulation, until spreading to the upper limbs and to the involuntary musculature [4,5]. From a cellular point of view, IAHSP is characterized by unorganized cell trafficking and neuronal development, associated with defective mitochondrial dynamics and generalized oxidative stress [6,7]. These pathologic features of IAHSP are common to ALS II, JPLS, and other MNDs [1,3,4].

Key events responsible for those MNDs are mutations of the gene *ALS2*, encoding for the cell trafficking-related protein alsin. Thus, understanding the molecular mechanisms of alsin is of extreme interest for unraveling the pathogenetic mechanisms of IAHSP and several other MNDs [3]. Clinical symptoms are observed with mutations occurring in compound heterozygosis (or homozygosis) [1,4], meaning that, in principle, correcting for the effect of one mutant allele might suffice to treat IAHSP patients.

Alsin is a 184-kDa protein of 1657 amino acid residues (UniProt code: Q96Q42). The sequence includes three domains, which are essential for the activity (Figure 1A): the N-terminal regulator of chromosome condensation 1-like domain (RLD), the central Dbl-homology and pleckstrin-homology domain (DH/PH), and the C-terminal vacuolar protein sorting 9 (VPS9) domain. The RLD spans residues 1–690, with a β-propeller helical structure, composed of 7 blades. The structured component of this domain is interspaced by an intrinsically disordered region (IDR) of 307 amino acids [8,9]. Experimental evidence supports that RLD works as a molecular switch: in a steady state, it keeps alsin in the cytosol; upon activation, it operates a conformational change, driving the protein to localize to the endosome membrane compartment [9,10]. DH/PH is comprised in the region 691–1026 and evidence indicates that this domain mediates the interaction with the active form of RAC1 [11], event triggering the cascade that, lastly, leads to guanine nucleotide-exchange factor (GEF) activity to the endosome fusion-inducing factor Rab5 through VPS9. This domain, located in the C-terminal region of alsin (1392–1657), is not just responsible for Rab5 GEF activity, but also works as key interaction site for homolog dimerization of two alsin units [10]. Between DH/PH and VPS9, there are a series of repeated motifs (MORN repeats), with regulatory functions, even though still poorly characterized [10].

Certain aspects of alsin’s function are experimentally known: alsin can form oligomers in the cytosol, and specifically, its most abundant physiologic form is a tetramer. In a previous work, we elucidated a possible tetramer structure [12]. This mutimeric state is hypothesized, allowing for the correct recruitment of active RAC1 and other oligomers that result in altered function [9]. The tetramer is first assembled by the homolog interaction of two VPS9 domains, followed by the interaction of two dimer units through their DH/PH regions. The formation of the tetramer is linked to endosome localization and Rab5 GEF activity. Even if not directly involved in the tetramerization, RLD is necessary for the correct regulation of endosomal localization of alsin. The current interpretation is that this happens through a conformational change [10]; thus, structure integrity and a prior correct assembly of two alsin monomers are crucial for the activity.

Mutations reported in the literature are coherent with truncated alsin forms, reputed to be degraded and depicting a loss-of-function pathogenesis. However, some patients report missense mutations, leading to protein products. In practice, there are few IAHSP patients, but each of them has specific missense mutations. The first challenge that such a broad mutational landscape offers is that the different mutations correspond to different multimers and, thus, to different pathogenetic mechanisms. Individual treatment is, therefore, needed for any patient presenting different pathological alsin mutations.

In this paper, we focus on a patient case (OA) harboring two mutations in the C-terminal region: one allele translates a frame-shifted, truncated form of alsin, while the other one harbors a R1611W substitution (in VPS9) [13]. As already discussed in a previous report [12], we repute the second mutation as potentially druggable: we focused our efforts on this mutant.

With the aid of in silico tools and cell-based experiments, the main aims of this work are: (a) to characterize and verify the relative stability of physiological and pathological dimerization modes obtained from the 3D structure of normal and mutated VPS9, (b) to repurpose already commercialized drugs to correct the pathogenic mechanism, and (c) to perform proof-of-concept experiments testing the best identified hit (Figure 1B).

Overall, this paper is expected to provide a potential drug candidate for the treatment of a specific IAHSP form and, further than that, shed light on alsin’s function, a key mechanism in the pathogenesis of several motor neuron diseases, such as ALS. Moreover, by setting-up a strategy to establish a mutation-based drug discovery pipeline and verify its validity, we expect to demonstrate the potential of application on other cases.

## 2. Results and Discussion

### 2.1. Setting-Up the Computational Strategy

The schematic description of the research strategy is in Figure 1B. A literature search showed that there is no crystallographic structure of alsin. However, our patient expresses a R1611W-mutated VPS9 domain, and previous experimental data suggest that knowing the 3D structure of VPS9 domain and RLD is sufficient to characterize the impact of the mutation on the tetramerization process [10]. This information allows us to focus on 3D models of each involved domain, avoiding working with the whole protein, which is extremely demanding, in terms of computational power.

The homology-modeled structures of VPS9 and RLD were obtained and validated with alsin’s model [12], released with AlphaFoldDB [14]. Then, we applied a pool of innovative methods to verify the impact of R1611W mutation on homo- (VPS9/VPS9) and hetero- (VPS9/RLD) dimerization through structural and energetic criteria. Analogs, two additional VPS9 models harboring mutants P1603A and L1617A with known activity [15], were generated and used to validate our results about R1611W.

Then, we modeled the interaction complex between the human Rab5 and VPS9 domains by employing a crystal structure as template. We used the model to check whether residue 1611 participates in the Rab5/alsin interaction, which is likely to impact the GEF function.

Finally, we performed a virtual screening to find a drug candidate binding the mutant residue, thus allowing for dimer and tetramer formation.

Proof-of-principle cell-based experiments were used to validate our computational findings.

### 2.2. 3D Models of VPS9 and RLD Domains and Related Mutants

First, we computed a reliable 3D homology model of alsin’s VPS9 domain. As already shown by Del Prato and coworkers [16], the majority of the commonly used alignment algorithms fail to properly locate the gaps when aligning the whole VPS9 sequence to resolved templates. We, therefore, aimed to perform a three-step modeling by subsequently predicting the 3D structure of the following regions: VPS9 core domain (residues 1513–1649, containing the Rab5-GEF region), helical bundle (HB, 1392–1512), and lastly, the 20 aa linker region (Figure 2A). The VPS9 core was modeled as previously described [12]. The HB region was modeled based on the PDB 2OT3. The linker sequence connecting the VPS9 core to HB used PDB 3LP8 as a template. Further details are in the methods section and Figure 2A. The final 3D models (WT and R1611W) spanned positions 1392–1657 (Figure 2B). Analysis of the Ramachandran plot revealed a limited number of violations, agreeing with plausible structural features (Figure 2C).

Modeling the mutation from OA (R1611W) followed the same procedure. Moreover, we built two additional VPS9-HB models, also impacting on oligomerization and based on the findings from some of us [10,15]. Those are: P1603A (tetramer-capable, Rab5-GEF defective) and L1617A (both tetramer and GEF incapable). Further details are in the Appendix A.

Regarding the N-terminal RLD, we obtained the 3D model from the lab of Prof. R.S. Devon, as reported in the publication from Soares and co-workers [8] (Figure 2D).

### 2.3. VPS9 Models Stability

The first refinement to the definitive VPS9-HB models (residues 1392–1657, from now on referred to as VPS9) was performed by submitting them to a minimization cycle with MOE (see Methods, Section 4) and observing the variation of their atomic coordinates. This aspect gives the first information about the stability of a 3D model, in terms of closeness to a minimum energy conformation. Indeed, both models showed sufficiently low RMSD values when superposing them to their minimized structures (Figure 3A), supporting their reliability.

We aimed to further investigate the stability of our models by submitting them to a flexibility analysis with CABSFlex2.0 [17]. This freely online available tool is based on a coarse-grained molecular dynamics simulation, and it is able to identify flexible portions of globular proteins. The scope of this analysis is two-fold: (a) it suggests which residues are more prone to de-stabilize the protein secondary structure, and (b) it offers a useful comparison to understand whether mutation R1611W affects VPS9 stability. Except for the linker region (around residue 1500) showing some variability, the rest of VPS9 showed comparable flexibility and stability (Figure 3B). The area of the mutation results was slightly more rigid (Figure 3B, black arrow), as also confirmed by normal mode analysis with WebNM 2.0 [18] (Appendix A).

Overall, these results suggest that the pathogenesis might derive from the different physicochemical properties of the mutated residue, more than from a structural re-arrangement.

### 2.4. Homo and Heterodimers: Building and Structural/Energetic Analysis

We have previously shown that VPS9 is essential for the tetramerization and that mutations to this domain mainly impact the aggregation state [10]. In turn, alsin constructs missing RLD can tetramerize and localize to membrane ruffles without RAC1 activation, preferentially localizing to early endosomes, although uncontrollably activating Rab5. One exception is constituted by mutation R1611W: even though affecting VPS9, this mutant shows both abnormal oligomers and impaired endosome localization [10], suggesting a potential impairment of the RLD regulatory function. We, therefore, hypothesized that mutant VPS9 has abnormally high affinity for RLD. The foundations of this hypothesis are that: (1) RLD is prone to hydrophobic interactions [10], and (2) a basic, charged arginine is substituted by a lipophilic tryptophan, a newly acquired hydrophobic site.

To this aim, we first submitted the VPS9 core structure to a dimerization potential analysis with ZDock [19]. This protein docking engine generates a visual report, highlighting the center of mass of the top 500 dimers obtained from the analysis. When comparing the wild-type VPS9 to the R1611W-mutated variant, a newborn interaction cluster appears in correspondence of the mutated residue (Figure 4A).

In order to accurately predict the probable dimer models, we submitted a query of VPS9 domains to the protein docking engine ClusPro [20] with either a second unit of VPS9 or RLD as interactor (Figure 4B). This docking engine utilizes a fast Fourier’s transform, allowing the sampling of a high number of orientations and, thus, obtaining refined dimeric poses. A series of 30 dimer structures was returned, ranked by probability (Appendix A). A first visual inspection analysis of the docked dimers highlighted that mutated VPS9 tends to internalize residue 1611 at the interaction interface more than in the WT counterpart, especially in the VPS9-RLD dimers, as highlighted by the rank-weighted externalization score (Figure 4C) and the representative structures (Figure 4D).

We have previously reported that RLD has an extremely hydrophobic surface at the “bottom face” of its β-propeller structure [10]. We verified by visual inspection that this is the most favored interaction site for mutant tryptophan 1611 (Figure 4E). Calculation of the solvent-accessible surface area (SASA) of residue 1611 highlighted lower SASA values for VPS9-RLD dimers (Figure 4F), suggesting that the strong solvation penalty for tryptophan 1611 might drive the formation of a more stable complex between the mutated VPS9 and RLD.

To further support our hypothesis, we performed a calculation of the interaction energy (see Methods, Section 4). The result of this analysis revealed a trend of preferential interaction between mutated VPS9 and RLD (Figure 5A).

Furthermore, we aimed to test the stability of the VPS9/VPS9 and VPS9/RLD dimers by calculating the free energy differential ΔΔG with the recent model developed by Caldararu and colleagues [21] (Figure 5B, see Methods, Section 4). This method interpolates the variation of free energy upon simulated amino-acid mutation, with a linear combination of 2D/3D molecular descriptors. Analysis of residue 1611 can be utilized as a method to test the stability of this interaction. In agreement with the previous data, the substitution of residue 1611, bound to RLD, revealed the lowest ΔΔG values (Figure 5B). This highlights, once more, the higher stability of this complex and, therefore, suggests that R1611W pushes VPS9 to prefer RLD over another VPS9 as an interaction partner.

We wanted to verify our findings with a more mechanistic approach, thus we submitted the top-score docked dimers (see above) to a steered molecular dynamics simulation (SMD). A molecular dynamics simulation (MD) reproduces the behavior of molecular systems in a defined timespan and calculates the atomic coordinates evolution under controlled thermodynamic conditions. The difference between a classic MD and SMD is that the latter applies restraints to a defined subset of atoms (anchor) and contemporarily simulates a pulling force applied to another subset (pulling). Accordingly, when pulling it apart, the later the complex dissociates, the higher the affinity.

Analysis by global RMSD is inconclusive (Appendix A), while focusing on residue 1611 shows that tryptophan 1611 is one of the last residues detaching from the surface of RLD (Appendix A). A visual inspection of the SMD frames extended this consideration to the whole VPS9 core (Figure 5C and Appendix A): residues of the R1611W VPS9 core lag the RMSD increase, revealing stronger interactions and being less prone to detach (Figure 5D). These data are corroborated by the RMSD of residues composing the VPS9/HB linker, displaying systematically higher values for mutated VPS9, due to stretching (Appendix A). Conversely, WT VPS9 immediately breaks its interaction with RLD. The major affinity of R1611W VPS9 for RLD is statistically supported by average RMSD values (Figure 5E).

Taken together, these data confirm that the mutation to tryptophan in position 1611 makes the VPS9/RLD interaction more favored, most likely favoring the abundance of this complex.

### 2.5. VPS9 and Rab5

Previous data highlight that mutant VPS9 prefers RLD as interaction partner, but still the question remains whether the R1611W mutation could affect the intrinsic GEF capacity of VPS9 too. To this aim, we constructed a VPS9-Rab5 interaction model by utilizing the co-crystallized structure of Rab5 (*A. thaliana*) in complex with Rabex5-VPS9 as a template (PDB 2EFC, Figure 6).

The results corroborate the hypothesis that the mutation solely leads to incorrect dimerization, without affecting GEF function. In fact, arginine 1611 is located on the opposite face, respectively to Rab5 (in green, Figure 6A), and not involved in the interaction. Moreover, the key residues individuated by Del Prato and colleagues [16], essential for the GEF activity of VPS9 domains, are conserved in alsin, not interacting with residue 1611 (Figure 6B). Taken together, these findings support that mutation R1611W does not affect the intrinsic GEF capability for Rab5. This hints to the potential of a pharmacologic strategy aimed to mask tryptophan 1611.

### 2.6. Analysis of Additional Mutants

Alsin mutations have been previously investigated by some of us [10,22]. Besides R1611W, P1603A and L1617A are also of particular interest for this study. Proline 1603 is one of the Rab5 GEF functional conserved residues described by Del Prato and colleagues [16], and it is present at the interface between Rab5 and VPS9, far away from residue 1611 and not interacting with RLD, according to our models (Figure 6A). Its function is, therefore, just related to the GEF activity. Indeed, P1603A is described as tetramer-forming, endosome-localizing, lacking just Rab5-GEF activity [10,23]. Mutation L1617A (artificially introduced for research purposes) [15] is, on the other hand, not able to form proper tetrameric structures in the cytosol (as R1611W).

As further verification, we modeled the VPS9 structures harboring mutations P1603A and L1617A, submitted them to docking with ClusPro, and tested the top-score dimers with SMD, as performed for R1611W. P1603A provided a straightforward interpretation: the mutant behaves closest to WT, with analog RMSD values (Appendix A). A different discussion must be had for mutation L1617A. Previous experimental results [10] suggest a potentially similar dimerization with RLD, as for R1611W (Figure 5C,D), but this was not verified in our SMD simulation: the interaction between L1617A and RLD reflects comparable strength to the one of WT (Appendix A). Moreover, L1617A shows higher affinity for VPS9 (Appendix A). This occurs due to a different preferred dimerization mode of the two VPS9 units (Appendix A), which is, in our interpretation, the reason why L1617A is also not able to form tetramers.

Taken together, these data reveal important mechanistic aspects about non-tetramerizing VPS9 mutations: alsin forms abnormal homodimers (Appendix A), impairing the formation of tetramers.

### 2.7. Virtual Screening to Identify Small Molecules Restoring the Functionality of Alsin

The verification that R1611W can influence tetramerization, rather than the Rab5-GEF mechanistic directly, suggests that masking it could re-establish the correct tetramer formation, without affecting the Rab5-GEF activity. We, therefore, envisioned that the next step of our study should be the identification of small molecules binding the mutant tryptophan and restoring the preference of VPS9 for the homolog interaction (Figure 7A).

To this aim, we looked for a binding pocket near R1611W: nine residues within 5Å were identified (P1552, V1556, Y1607, V1608, V1609, L1610, W1611, A1612, and R1613) and submitted to a pocket definition with three different tools (PocketQuery, PrankWeb-CavityPrank, and CASTp 3.0, see Methods, Section 4). In agreement, a unique pocket was identified (Figure 7B). This is of crucial importance, since the screening of small molecules on exposed surface sites often results in low affinity and specificity.

The pocket was submitted to a structure-based virtual screening of 9815 approved molecules, retrieved from ZINC (v.v. 15) and Drug Bank (GOLD software, see Methods, Section 4). The initial screening outputs were filtered with blood–brain barrier (BBB) permeability in silico models (see Methods, Section 4). The top hit was menaquinone 5 (MK5). A subsequent screening refinement procedure identified menaquinone 4 (MK4, Figure 7C), which was selected as the most promising candidate. MK4 was approved for the treatment of osteoporosis in Japan^25^ and commercialized as supplements in Europe and USA. The top-ranked docking pose shows specificity for the hydrophobic tryptophan in position 1611 (Appendix A), with a PLP fitness score of 78, while the best WT pose stops at 52 points (the higher, the more specific, Appendix A). This trend is confirmed by the top 10 best docking PLP fitness scores (Appendix A).

Finally, we verified the specificity of MK4 for R1611W with a different docking method, AutoDock Vina [24]. This analysis provided an affinity difference of −13.3 kcal/mol, predicting that MK4 preferentially binds R1611W and not the WT residue. Since the VPS9 domains are highly conserved in several non IAHSP-related proteins, our data are a good, bona fide indicator of the low chance that MK4 has off-target effects by binding other WT VPS9 domains.

### 2.8. Experimental Validation

To confirm the in silico predictions, we performed biochemical analyses in the presence/absence of MK4 (Figure 8).

Cell lysates prepared from HeLa cells, expressing either wild-type alsin or the R1611W variant, were subjected to gel-filtration chromatography. Wild-type alsin (184 kDa) was eluted in fractions with an apparent molecular mass of 600~700 kDa, regardless of (a) the duration of the MK4 treatment (for 4 h and 24 h, Figure 8) and (b) the presence of MK4. R1611W was broadly eluted in fractions, with apparent molecular masses ranging from 300 to 700 kDa, plus a minor peak at ~1000 kDa (Figure 8A,B). These data support that R1611W exists as dimers, trimers, and aberrant high-molecular weight aggregates [10]. Remarkably, the 4 h treatment with MK4 resulted in an increased proportion of tetrameric R1611W complex. Furthermore, the 24 h treatment not only decreased the levels of high-molecular weight aggregates, but also increased the tetrameric forms, although the longer-period expression of R1611W (24 h) itself led to the increased levels of aberrant high-molecular weight aggregates (Figure 8C,D). These results indicate that MK4 can effectively restore the aberrant R1611W into the normal structure of the complex.

Finally, our preliminary immunocytochemical analyses showed that the treatment with MK4 slightly increases the endosomal localization of R1611W (Appendix A) and decreases the formation of alsin-localizing aggregates in cells (Appendix A).

## 3. Conclusions

Without knowing the molecular basis of alsin-mediated action, no therapeutic treatment for *ALS2*-related disorders can be found. Molecular modeling and virtual screening are low-cost computational tools that can (a) unravel the molecular interactions governing alsin oligomerization and (b) discover compounds to restore alsin’s functionality.

Based on our results, we hypothesized that R1611W alsin preferentially interacts with RLD through the hydrophobic indole moiety of the mutant tryptophan. This blocks the formation of tetramers and stops RLD from correctly undergoing the necessary conformational changes required for its function. The preferred dimerization mode analysis and interaction energy calculations support this hypothesis. This model explains the RLD-dependent mislocalization to endosomes upon RAC1 activation of VPS9-mutant alsin.

As result of the virtual screening, we individuated MK4: it can specifically bind to the mutant tryptophan in position 1611, shielding the hydrophobic moiety and re-establishing the correct VPS9/VPS9 dimer, tetramer, and physiologic functions. The importance of repurposing MK4 for R1611W is supported by the fact that this molecule is already approved for therapy and has demonstrated safety in pediatric patients [25].

Notably, vitamin K derivatives display anti-oxidant properties [26,27,28]. The body of scientific literature exhaustively reports that several MNDs are characterized by oxidative stress [6] and other defects in the mitochondrial compartment [29]. Metabolically stressed cells tend to fission their mitochondrial network [30] as a protecting mechanism against free radical production [31]. This requires supplying membrane portions to the mitochondrial network, and this happens with a Rab5-depending mechanism [31,32]. The work of Hsu and colleagues [32] links defective fission to alsin’s function, suggesting a contribution of tetramer-incapable mutant alsin to the oxidative stress increase. Thus, the use of vitamin K derivatives could have a two-fold mechanism of action: (1) restoring the tetramerization and (2) as anti-oxidant [26,27,28].

In this work, we unraveled the molecular mechanism of one specific alsin mutation, known for originating IAHSP. Further than that, we found and tested a drug already approved in Japan, which our experiments support as re-establishing alsin’s function (for mutant R1611W) and mitigating the oxidative damage. Nevertheless, our conclusions are proofs of concepts, based on early and preliminary data; thus, we need to remain cautious about the extent of the effects here observed, and further investigation is necessary to strengthen our conclusions. In our view, the next step to this aim should involve testing vitamin K derivatives on more relevant-to-human models, such as induced pluripotent stem cells (iPSC). Obtaining such a model is of great translational value, both as a drug testing platform and to further investigate the molecular mechanisms of ALS2-related MNDs.

Despite some of the above-mentioned limitations, our work was enough to obtain the approval of MK4 for compassionate use to treat OA, and the preliminary results are quite encouraging. This reveals that a strategy coupling protein modelling to virtual screening and experimental verification could be extremely valid for addressing similar cases, too. Specifically, the application of computational techniques to repurpose safe and already approved compounds for single cases is a core strategy for reducing the duration and costs. However, we would like to stress that computational predictions should always be experimentally validated and integrated in an experimental framework to lead to solid conclusions. Our strategy for this case represents an always perfectible pipeline, which we intend to personalize and apply to other patients suffering from rare genetic diseases.

## 4. Materials and Methods

Sequences and homology modelling: The original sequence of alsin was retrieved from UniProt (www.uniprot.org, accessed on 9 March 2022), gene name, *ALS2*, Uniprot ID: Q96Q42), and exported as FASTA sequence. The recognition of VPS9 core, HB, and linker was performed by following the alignment with human Rabex5 sequence, published by Del Prato and colleagues [16]. The VPS9 core (residues 1513–1657) was modelled (as previously shown [12]) with the module of Modeller [33] (salilab.org/modeller/, accessed on 9 March 2022) integrated in Chimera [34] (www.cgl.ucsf.edu/chimera/, accessed on 5 March 2022), with standard settings. The selected 3D templates were PDB 2OT3 and 1TXU.

The HB sequence (residues 1392–1512) was modelled utilizing the homology model wizard built-in to the Molecular Operating Environment (MOE, www.chemcomp.com, accessed on 5 September 2022) by taking PDB 2OT3 as unique template. A superposition of both models with the crystal structure of human Rabex5 from PDB 1TXU provided an adequate spatial alignment to model the linker between the two models.

The linker sequence ALDNDREEDIYWECVLRLNK (residues 1493–1513), spanning from the first α-helix of the HB to the first N-terminal residue of the VPS9 core sequence, was modelled with the MOE linker modeler wizard tool by using the PDB 3LP8 as template. The force field parameters utilized for local and global minimizations are AMBER 12:EHT, implicit solvation, and R-field, as prescribed by the manufacturer’s tutorial for globular protein modeling.

The P1603A and L1617A VPS9-HB models were obtained by mutation of the WT VPS9-HB model with the Chimera built-in tool and sidechain optimization tool [34].

Flexibility assessment: The obtained models were first submitted to a RMSF (root mean square fluctuation) analysis, based on a short coarse-grained MD run, with the webserver CABSFlex2.0 [17].

The effect of mutation R1611W was confirmed by normal vibrational mode analysis performed with WebNM [18]. Specifically, the comparative analysis was chosen by uploading the WT model and simulating the mutation of arginine in position 1611 to tryptophan.

Dimer poses and analysis: the dimer interaction potential analysis was performed with ZDock [19] (https://zdock.umassmed.edu/, accessed on 20 March 2022) with standard settings. This tool simulates the docking of the two monomers and outputs a map of potential contacts in the space surrounding the structure.

The generation of definitive dimer poses was performed with ClusPro [35] (https://cluspro.bu.edu, accessed on 20 March 2021) on a CPU server with standard settings. As with other force field-based docking methods, ClusPro performs its task in three steps: (1) sampling billions of conformations and rigid-body docking poses, (2) selecting the first 103 lowest energy conformers, and (3) optimizing the steric clashes and energy minimization. The particularity of this method is the application of a specific fast Fourier’s transform in performing step 1, allowing faster and more reliable calculations. Indeed, ClusPro has been reported to outperform other protein docking methods (such as ZDock), becoming the gold standard in recent years [20,35].

The externalization analysis of residue 1611, as well as the interaction analysis with RLD bottom face, was performed by visual inspection of the structures in Chimera [34]. Residue 1611 was highlighted, and a boolean score was assigned to the structure (internal/external). The score was then converted in 1 or 0 and weighted by the energy output scoring obtained from ClusPro. The results for each condition are expressed as means in the bar charts, and error bars depict standard deviations.

Dimers stability and SASA calculation: The interaction energy calculation was performed with the force field (FF) AMBER 12:EHT, implicit solvation model R-field, and ε = 80, as implemented in the Schrödinger suite (version 2021-1, www.schrodinger.com, accessed on 25 March 2022). Each dimer structure was imported as PDB, hydrogen atoms were added, charges resolved according to FF parameters and free energy of the complex, and each separate monomer was calculated. The difference between total system energy and the sum of each single term was considered an interaction energy estimation.

The calculation of ΔΔG was computed with the method described by Caldararu and colleagues [21]. In this case, the best performing model, SimBa-NI, was utilized. The three descriptors, on which is regression model was built, are, respectively, Vdiff (difference in average volume upon residue mutation), Hdiff (enthalpic contribution of the mutation), and SASA (solvent-accessible surface area). The first two descriptors, being 2D, were calculated upon the tables provided with the original publication. The last descriptor was computed with FreeSASA [36], a solvent-accessible surface area (SASA) calculation tool run on a WSL-Ubuntu virtual Linux environment. The only parameters that were changed from standard in FreeSASA were: solvent radius = 1.4; input: naccess, in agreement with the original publication of Caldararu and colleagues [21].

Steered molecular dynamics (SMD) simulations were performed in NAMD [37], version 2.13, with CUDA implementation and CHARMM36 FF, with the following protocol: the top hit dimer structures obtained from ClusPro [20] protein docking were opened in Chimera [34], and the residue coherently renumbered, in order to avoid redundancy. The obtained structures were exported as PDB files and imported in the VMD [38] plugin QwikMD [39] for SMD setup with the following parameters: solvent = explicit; salt (NaCl) = 0 (neutralization of the system); equilibration = y; MD = n; SMD = y; temp = 27 °C (300 K); pulling distance = 50 Å; pulling speed = 10 Å/ns; pulling residues = “all residues from one VPS9”; anchoring residues = “all residues from either one VPS9 or RLD”; live view = uncheck. The configuration files were exported, and the simulations were run on a workstation with the following specifics: core = XeonXX; RAM = 32GB; 2TB SSD; GPU = Titan XP quad-core; OS = Linux CentOS vv 7. Analysis of the trajectories was performed in VMD [38], version 3.1, and the dedicated RMSD trajectory plugin.

Pocket definition and Virtual screening: The VPS9 residues with heavy atoms within 5 Å from W1611 were highlighted with Chimera [34] and submitted to pharmacophore identification with PocketQuery [40] (http://pocketquery.csb.pitt.edu/, accessed on 13 April 2022). The cavity identification was performed with PrankWeb-CavityPrank [41] (http://prankweb.cz, accessed on 13 April 2022) and confirmed with CASTp 3.0 [42] (http://sts.bioe.uic.edu/castp, accessed on 13 April 2022) and ProteinPlus [43] (https://proteins.plus/, accessed on 13 April 2022).

Identifying a cavity allowed us to define a query for the virtual screening (VS). The pool of screened molecules consisted of already therapeutically-available molecules: 992 were from ZINC15 [44] and 8823 from DrugBank [45] databases. The vs. was performed with GOLD [46] (www.ccdc.cam.ac.uk, accessed on 1 May 2021) version 2020.2.0. The cavity parameters: center on Trp99 (W1611); point: 15.433, 6.723, 5.661; rest: default parameters in the wizard. Fitness scoring function: ChemPLP. Scoring parameter: Goldscore.

The vs. refinement was performed by selecting top hits (Goldscore greater than 70) and re-submitting them to search with scoring function ChemPLP.

The docking of MK4 with AutoDock Vina was performed with the web interface Webina [24], by submitting protein and ligand in pdb format.

The blood-brain barrier permeability and other ADME parameter predictions for the top screening hits was performed by importing the results in VolSurf+ [47] and calculating the following descriptors: lgBB, SOLY, LOGP n-Oct, and LOGP c-Hex, as prescribed by the developer (www.moldiscovery.com, accessed on 1 May 2021).

Graphical representation and statistical analysis: Protein schemes were drawn with the free package Illustrator for Biological Sequences [48]. Graphs and statistical analysis were drawn and calculated with GraphPad prism, version 6.0 (http://www.graphpad.com/, accessed on 1 June 2022). Pictures of 3D structures are from Chimera [34] and VMD [38].

Gel-filtration chromatography: HeLa cells were cultured in Dulbecco’s modified Eagle’s medium with high glucose (DMEM-HG; Wako) supplemented with 10% heat-inactivated fetal bovine serum gold (PAA Laboratories GmbH), 100 U/mL penicillin, 100μg/mL streptomycin, and 1 mM sodium pyruvate. HeLa cells were transfected with pCIneoFLAG-ALS2^WT^ or pCIneoFLAG-ALS2^R1611W^ expression constructs [10] using the Effectene transfection reagent (QIAGEN), according to the manufacturer’s instructions. Twenty-four hours (hr) after transfection, the cells were treated with 20 μM MK4 (Sigma Aldrich, Menaquinone: MK2, catalog n: 47774, CAS number 863-61-6) for either 4 h or 24 h. Then, the cells were washed with PBS(−) and lysed in gel-filtration buffer (50 mM Tris-HCl (pH 7.5), 150 mM NaCl, 0.1% IGEPAL CA-630 (SIGMA), and complete protease inhibitor (Roche Diagnosis)). Supernatants were recovered by centrifugation at 23,000× *g* for 15 min and loaded onto the column (Superose 6 10/300 GE Health Life Care), and the eluates were fractionated and collected. Elution was conducted at 4 °C, at a flow rate of 0.3 mL/min, with a fraction volume of 0.5 mL. Fractionated samples were analyzed by Western blotting using the anti-ALS2-MV (MPF1012-1651) polyclonal antibody, as previously described [49]. The elution profile of the column was calibrated with the size standards (IgM, 975 kDa; thyroglobulin, 669 kDa; ferritin, 440 kDa; aldolase, 158 kDa; and ovalbumin, 43 kDa). Signals were visualized by immobilon (Merck) and detected by EZ capture MG system (ATTO). Signal intensities of Western blots were quantified by CS Analyzer, version 3.0 (ATTO).

Immunocytochemistry: HeLa were transfected with pCIneoFLAG-ALS2^WT^ or pCIneoFLAG-ALS2^R1611W^ expression constructs [10], with or without pCIneoHA-Trio-GEFD1 (1233–1628) [50]. Six hours after transfection, the cells were treated with 10 μM MK4 and cultured for additional 24 h. Then, the cells were washed with PBS(−) twice, fixed with 4% paraformaldehyde (PFA) in PBS(−) for 20 min, and permeabilized with 0.1% triton X-100/5% normal goat serum (NGS) in PBS(−) for 30 min. Anti-ALS2-RLD (HPF1-680) [15] and anti-transferrin receptor (TfnR) antibodies, diluted in PBS(−) containing 0.05% triton X-100/1% NGS, were added to the fixed cells and incubated overnight at room temperature. After washing with PBS(−), samples were incubated with Alexa 488- or 594-conjugated anti-mouse or rabbit IgG (molecular probes) in 0.05% Triton X-100/1% NGS/PBS(−). After washing with PBS(−), the cells were mounted with VECTASHIELD with 4′,6-diamidino-2-phenylindole dihydrochloride (DAPI). We observed these cells by a fluorescence microscopy (KEYENCE BZ-710).

## Figures and Tables

**Figure 1 molecules-27-07063-f001:**
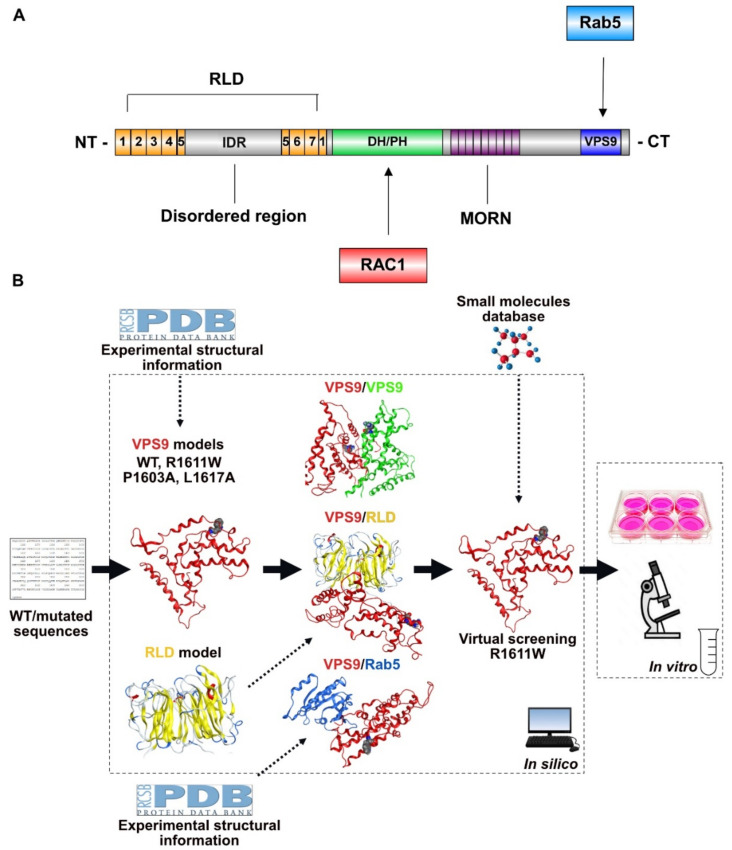
Alsin’s structure and our strategy. (**A**) Schematic representation of hitherto identified alsin domains and key interaction partners. (**B**) Schematic representation of our strategy. WT and mutant VPS9 models were built and homo-/heterodimerization modelled (VPS9/VPS9, VPS9/RLD, and VPS9/Rab5). Approved compounds binding mutant R1611W were identified with virtual screening and tested on HeLa cells.

**Figure 2 molecules-27-07063-f002:**
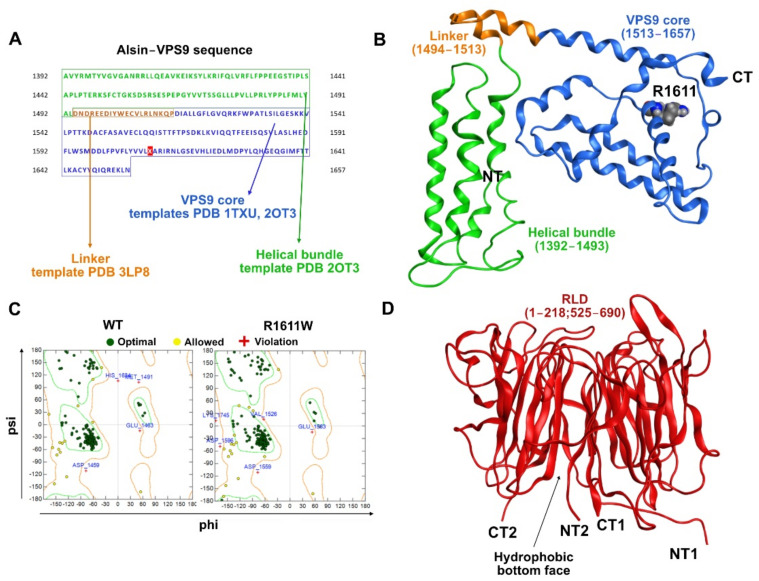
VPS9 and RLD models. (**A**) Sequence of VPS9 and modeling strategy or each portion. (**B**) Results of 3D homology modeling of VPS9. (**C**) Ramachandran plots of the modeled structures. (**D**) 3D structure of RLD from Soares and colleagues D) RLD model.

**Figure 3 molecules-27-07063-f003:**
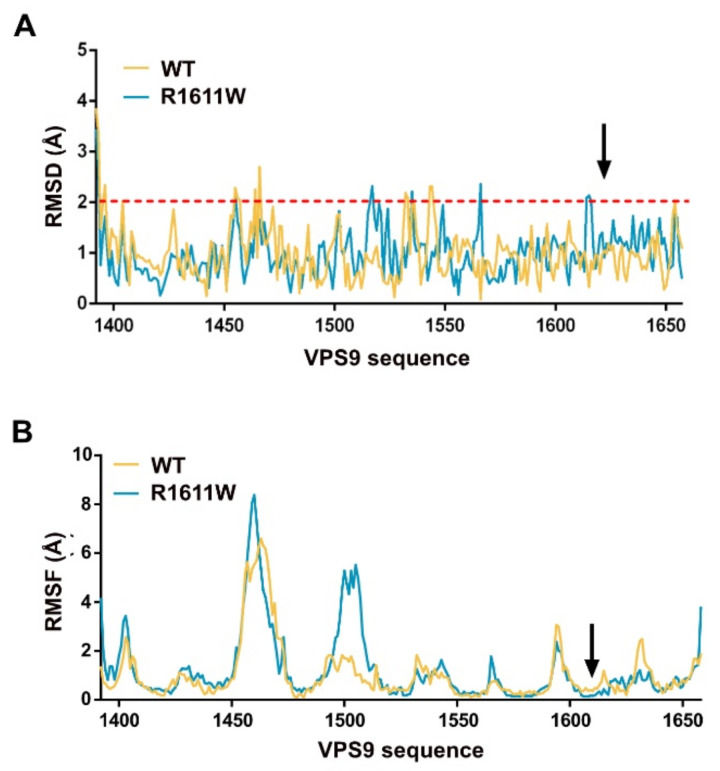
VPS9 stability profile. (**A**) Average RMSD of VPS9 residues after minimization. The red dashed line marks the threshold of commonly allowed RMSD to define structural similarity. (**B**) Flexibility analysis of VPS9 core structure performed with CABS-flex 2.0.

**Figure 4 molecules-27-07063-f004:**
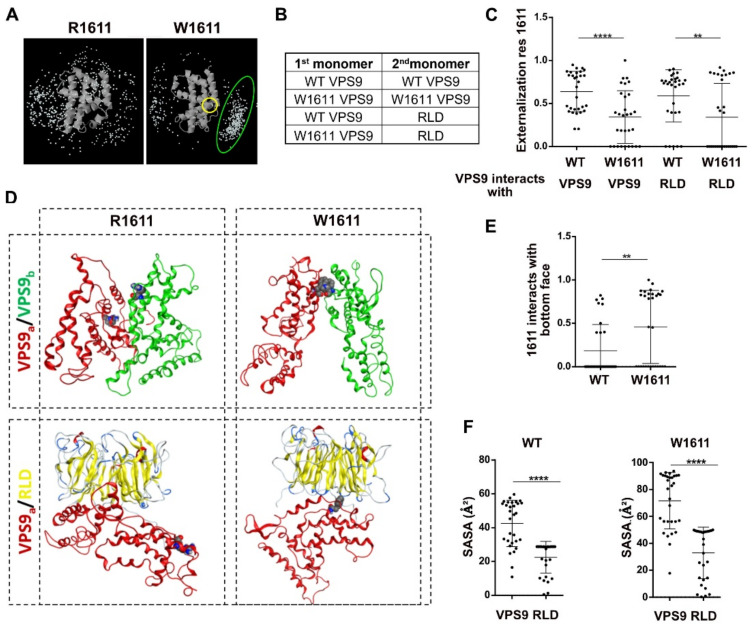
Interaction modes of WT and R1611W VPS9. (**A**) Interaction profiling of the space around VPS9 core. W1611 and its interaction cluster are, respectively, highlighted in yellow and green. (**B**) Scheme of submitted docking. (**C**) Externalization weighted score of residue 1611 in the dimers; each dot is a dimer pose; statistics: one-way ANOVA:(**D**) Representative structures from ClusPro docking study. (**E**) Interaction of residue 1611 with the RLD bottom hydrophobic surface; each dot is a dimer pose; statistics: unpaired *t*-test. (**F**) SASA of residue 1611 in the four complexes from (**B**); each dot is a dimer pose; statistics: unpaired *t*-test. *p*-value < 0.01 is **, *p* value < 0.0001 is ****.

**Figure 5 molecules-27-07063-f005:**
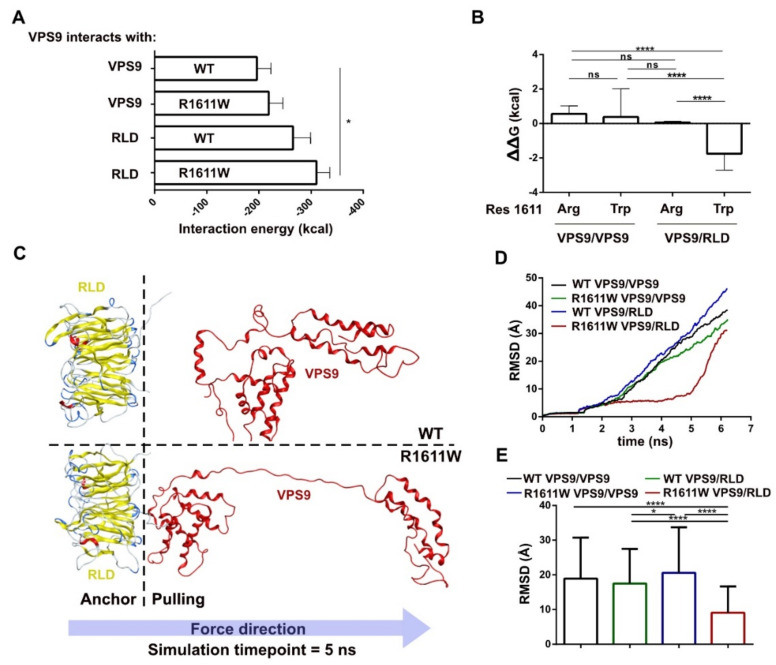
Dimers stability. (**A**) interaction energy calculated with AMBER. (**B**) ΔΔG calculation for dimers involving mutants of residue 1611; statistics: one-way ANOVA (**C**) Representative snapshot of steered molecular dynamics at 5 ns showing interaction of WT VPS9 (top) and W1611 VPS9 (bottom) interacting with RLD when VPS9 residues are pulled. (**D**) XY plot representing RMSD of VPS9 core over time in SMD from (**C**). (**E**) Average RMSD of VPS9 core throughout SMD production simulation from (**C**); statistics: one-way ANOVA. *p*-value < 0.05 is *, *p*-value < 0.0001 is ****.

**Figure 6 molecules-27-07063-f006:**
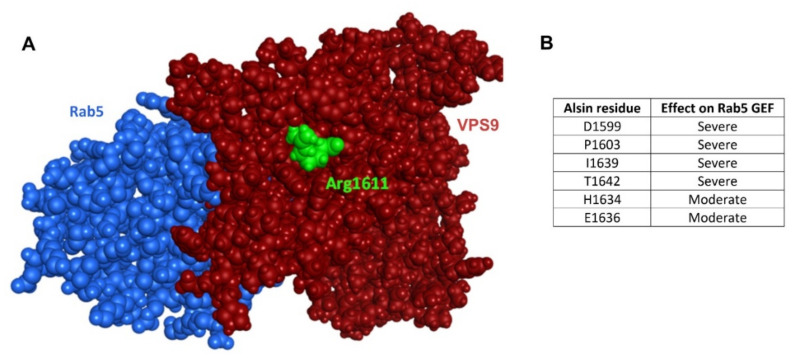
The interaction between Rab5 and VPS9 is not affected by residue 1611 mutation. (**A**) Interaction model of Rab5 (blue) and VPS9 (red); residue 1611 is highlighted in green and shown to be far from the protein interaction interface. (**B**) Table of alsin’s residues homolog to the ones individuated in human Rabex 5 as essential for performing GEF activity. They are, in large part, conserved in human alsin, and position 1611 is not among them.

**Figure 7 molecules-27-07063-f007:**
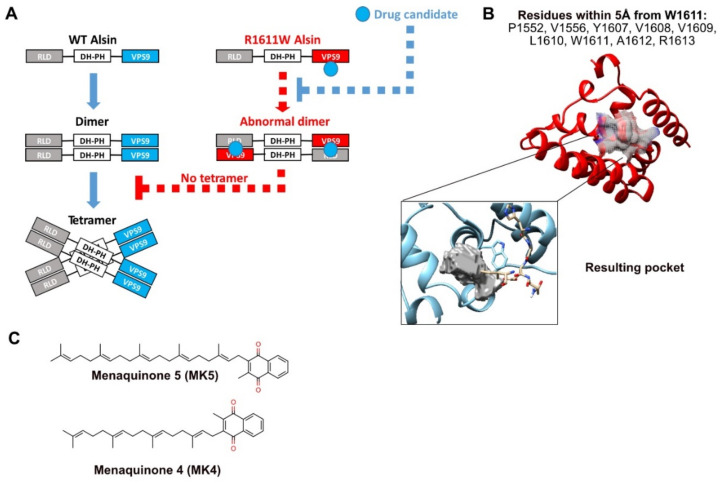
VS strategy and results. (**A**) Schematic representation of the hypothesized strategy scope of the VS. (**B**) Pharmacophore-residues surrounding W1611 and identified pocket. (**C**) Results of the VS: MK5 is the first result, MK4 from refined screening.

**Figure 8 molecules-27-07063-f008:**
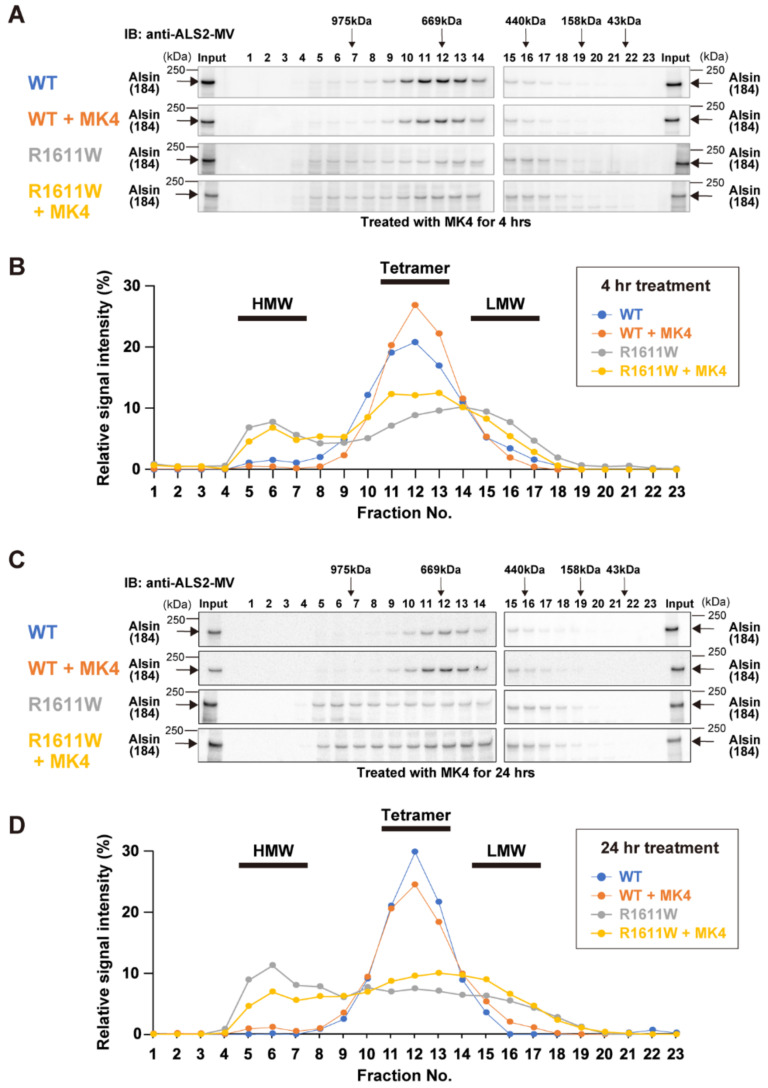
Gel filtration analysis of the wild-type alsin and R1611W pathogenic mutant. HeLa cells were transfected with pCIneoFLAG-ALS2^WT^ or pCIneoFLAG-ALS2^R1611W^. After 24 h, the cells were treated with MK4 for 4 h (**A**,**B**) or 24 h (**C**,**D**). Then, the cell lysates were applied to Superose 6 10/300 gel-filtration column. Cell lysates (indicated as input) and the fractions were analyzed by Western blotting using anti-ALS2-MV antibody (**A**,**C**). Arrows indicate the positions of molecular mass markers. (**B**,**D**) Quantitative analysis of signal intensities of alsin molecules. Each fraction was expressed as a percentage of the total amount of signals representing ALS2 across the entire fractions. (**B**,**D**) Positions of high-molecular weight (HMW) complex, tetramer, and low-molecular weight (LMW) complex are shown.

## Data Availability

The data presented in this study are available on request from the corresponding author.

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
