# Peer review of "Personalized Treatment for Infantile Ascending Hereditary Spastic Paralysis Based on In Silico Strategies"

_molecules, 2022, doi:10.3390/molecules27207063_

Round 1

Reviewer 1 Report

This is an original and timely approach in a rare inherited condition lacking therapies and with disease mechanisms poorly understood. The authors modeled the 3D structure of a protein whose crystal has not been resolved and defined new potential explanations as for disease mechanisms. Based on those studies, they attempted docking selection with possible FDA-approved drugs and picked two interesting compounds of which was later tested in terms of deleteriousness in cell culture. Overall this approach proved that combination of 3D modeling, in silico simulation and virtual drug screening can be helpful in a set of rare conditions where the disease mechanisms are hard to be understood.

Major comments:

1.    The authors should indicate in a separate paragraph the limitations of the study

2.    The authors should inform on the use of the two compounds (are they already in the market? What they serve for?) and advice cautiously on the very preliminary data in IAHSP to avoid easy misinterpretation or maybe misrepresentation

3.    Minor typos/spelling should be corrected

4.    A potential use of similar approaches in other conditions should be better emphasized

Author Response

Reviewer 1

  1. The authors should indicate in a separate paragraph the limitations of the study.

Good point. We discussed the limitations in the text accordingly. 

  1. The authors should inform on the use of the two compounds (are they already in the market? What they serve for?) and advice cautiously on the very preliminary data in IAHSP to avoid easy misinterpretation or maybe misrepresentation.

We thank the Reviewer for the comment. We added information about the use of K2 derivatives in the results section. Moreover, we now clearly state that we are discussing a single mutation, and advised caution in the interpretation of the results.

  1. Minor typos/spelling should be corrected

Ok, done.

  1. A potential use of similar approaches in other conditions should be better emphasized.

Ok. We stressed this aspect in the text.

Reviewer 2 Report

Sebastiano et al. provide in this study a potential of a pharmacologic strategy  to mask tryptophan 1611 in VPS9 to restore normal oligomerization properties of the R1611W aberrant protein. This aims for a personalized treatment for Infantile Ascending Hereditary 2 Spastic Paralysis.

In a first step the authors apply a comprehensive computational workflow to model the relevant protein complexes using online in silico tools including AlphaFold. After having demonstrated robustness of their models they performed a virtual screening to find a drug candidate from approved drugs set of ZINC binding the mutant residue to allow dimer and tetramer formation. The finally provided experimental results indicate that MK4  restores the aberrant R1611W into the normal complex structure.

The paper reads interesting and should be published.

The referee has two questions.

Line 191: This docking engine utilizes a fast Fourier’s transform, allowing the sampling of a high number of conformations and thus obtaining refined dimeric poses. Should the word conformations be replaced by the word relative orientations of the proteins?

348: These results indicate that MK4 can effectively restore the aberrant R1611W into the normal complex structure. Is it possible to quantify the effectiveness? From figure 8 it looks like the effect is minor?

Author Response

Reviewer 2

  1. Line 191: This docking engine utilizes a fast Fourier’s transform, allowing the sampling of a high number of conformations and thus obtaining refined dimeric poses. Should the word conformations be replaced by the word relative orientations of the proteins?

Ok, done

  1. 348: These results indicate that MK4 can effectively restore the aberrant R1611W into the normal complex structure. Is it possible to quantify the effectiveness? From figure 8 it looks like the effect is minor?

We thank the reviewer for pointing out this aspect. Although the experiment quantifies MK4-induced aggregation changes, we remain cautious about absolute quantifications: the experiment was conducted in a cellular model overexpressing Alsin, likely in a higher way than the physiological level. This proof-of-concept experimental setting is sided by another limitation: the concentration of MK4 can not be increased due to its solubility profile. Nevertheless, even though the observed effects were partial, we believe that MK4 is beneficial for the patient, at least to improve the disease symptoms.

The experimental limitations mentioned above explain the partial effect of MK4 in figure 8 and pointed out by the Reviewer. We strongly believe that this likely arises from the stoichiometric imbalance between an excessive number of Alsin molecules and a limited concentration of MK4 in cultured conditions. To quantify the effectiveness of MK4 on alsin’s tetramers, we would need to perform direct biophysical experiments based on purified Alsin and MK4. Such efforts are in due course in our laboratories.